Convergent origin of the narrowly lanceolate leaf in the genus Aster—with special reference to an unexpected discovery of a new Aster species from East China

Zhang Guo-Jin 1 2
Hu Hai-Hua 1 2
Gao Tian-Gang gaotg@ibcas.ac.cn 1 2
Gilbert Michael G. 3
Jin Xiao-Feng 4
1 State Key Laboratory of Systematic and Evolutionary Botany, Institute of Botany, Chinese Academy of Sciences , Beijing , China
2 University of Chinese Academy of Science , Beijing , China
3 Royal Botanic Gardens, Kew , London , United Kingdom
4 College of Life and Environmental Sciences, Hangzhou Normal University , Hangzhou , China
Culham Alastair
Electronic publication date: 2019 Jan 25
Publication date: 2019
Volume: 7
Electronic Location ID: e6288
Received 2018 Oct 10; Accepted 2018 Dec 15
Copyright: ©2019 Zhang et al.
Copyright year: 2019
Copyright holder: Zhang et al.
License: This is an open access article distributed under the terms of the Creative Commons Attribution License, which permits unrestricted use, distribution, reproduction and adaptation in any medium and for any purpose provided that it is properly attributed. For attribution, the original author(s), title, publication source (PeerJ) and either DOI or URL of the article must be cited.
License URL: https://creativecommons.org/licenses/by/4.0/

Keywords: BayesTraits, Astereae, Discrete data, Leaf shape, Riparian habitat, Preadaptation

Funding: National Natural Science Foundation of China 31870179 31570204 31270237 J1310002 S & T Basic Work 2014FY210300 2013FY112100 Sino-Africa Joint Research Centre SAJC201614 Strategic Priority Research Program of Chinese Academy of Sciences XDA20040000 This study was financially supported by the National Natural Science Foundation of China (No. 31870179, 31570204, 31270237, J1310002), S & T Basic Work (No. 2014FY210300, 2013FY112100), Sino-Africa Joint Research Centre (No. SAJC201614), and the Strategic Priority Research Program of Chinese Academy of Sciences (No. XDA20040000). The funders had no role in study design, data collection and analysis, decision to publish, or preparation of the manuscript.

==============================
Narrowly lanceolate leaves occur frequently in the genus Aster. It was often employed as a distinguishing character in the taxonomy of this genus. The origin of this particular leaf shape, however, has never been investigated using comparative methods. In this study, we reconstructed a comprehensive phylogeny that includes most species of Aster with narrowly lanceolate leaf. We then gathered data on riparian habitats and the presence or absence of narrowly lanceolate leaves, and investigated the evolutionary association between them in a phylogenetic context. Our analysis indicated that the species with narrowly lanceolate leaves are nested in unrelated lineages of the genus Aster, implying that they originated independently several times. Using Pagel’s comparative method of discrete data, we demonstrated a significant correlation between riparian habitats and narrowly lanceolate leaves. We further inferred the sequence of transition of the two characters. This analysis indicated that the sequence of evolution of riparian habitat and narrowly lanceolate leaf form was usually uncertain, but some positive results showed that the occurrence of riparian habitats may not precede the evolution of narrowly lanceolate leaf form. This study provided new insights into the adaptive evolution in a mega-diverse family. In addition, Aster tonglingensis, an unexpected new species with narrowly lanceolate leaves, was discovered and established based on the evidence from morphology, micromorphology and molecular phylogeny.

Introduction

How the environments modify morphology is one of the key questions in evolutionary biology (Grant & Grant, 2002; Lamichhaney et al., 2015; Malinsky & Salzburger, 2016; Meier et al., 2017). One focus is convergent evolution. Convergent evolution means that organisms from different lineages develop similar structures or forms in similar environments (Washburn et al., 2016). This phenomenon is widespread in plants, such as the lotus and water lilies, or the xeric highly succulent species of Euphorbia, Cactaceae and some species of Apocynaceae (McGhee, 2011; Alvarado-Cárdenas et al., 2013).

Asteraceae is a young family that originated at ca. 69–85 Ma (Barreda et al., 2015; Panero & Crozier, 2016). It is also the largest family of plants, containing nearly 30,000 species (Heywood, 2009; Funk et al., 2009). Members of this mega-diverse family occur in a variety of habitats, thus providing an excellent opportunity to study convergent evolution (Heywood, 2009).

Aster, the type genus of Asteraceae, contains ca. 150 species mainly distributed in Eurasia (Nesom, 1994; Nesom & Robinson, 2007; Chen, Brouillet & Semple, 2011). Its diversity centre is from East Asia to Himalaya (Chen, Brouillet & Semple, 2011). It occupies diverse habitats ranging from the Frigid Zone (e.g., Aster alpinus) to the Tropical Zone (e.g., A. philipinensis), from alpine talus (e.g., A. prainii) or alpine meadows (e.g., A. flaccidus) to forests (e.g., A. ageratoides) and coastal zones (e.g., A. spathulifolius). Some species occur in extreme dry hot valleys (e.g., A. poliothamnus) and others grow in wet places along the Yangtze River (e.g., A. moupinensis). The broad range and diversified habitats provide multiple niches and space for convergent evolution. For example, the short pappus, a character widely used in the taxonomy of Aster, was shown to be convergent (Ito et al., 1995). It has evolved several times within the genus Aster. Other characters with more plasticity, like leaf shape, however, have never been investigated in the genus.

Previous studies of the character of leaf shape in Ainsliaea (Asteraceae: Pertyeae) indicated that species growing in forests tend to have wide leaves which is of benefit to absorb sunlight (Mitsui et al., 2011; Mitsui & Setoguchi, 2012). In contrast, species growing along streams tend to have narrow leaves that can minimise any harm caused by water flow (Mitsui et al., 2011; Mitsui & Setoguchi, 2012). This narrow leaf shape of riparian species is an adaption to stream or river bank habitat. In the genus Aster, there are a few species with narrow leaves (e.g., Aster dolichophyllus Y. Ling, occurring as two small populations in Guangxi, China). To revise the genus Aster, we have conducted extensive field collections and observations in the field in Eurasia. During an expedition to south-eastern China in 2013, we encountered a distinctive species of Aster in Mt. Tongling National Forest Park in Wencheng county of Zhejiang province. It grew on rocks near a stream and had narrow leaves. In external morphology, it was very similar to A. dolichophyllus in having narrowly lanceolate, glabrous leaves and reflexed phyllaries. Similar morphology and habitat initially made us think that this plant might be conspecific with A. dolichophyllus. However, the distance between them was more than 1,000 km and there are many high mountains and big rivers separating the two places. So, our first question was: are these two Aster entities from these two distant places different populations of one species or are they two totally different species with similar morphology resulting from convergent evolution in similar habitats? There are also other species in Aster with similar narrowly lanceolate leaves and/or growing in riparian habitats, such as A. moupinensis (Franch.) Hand.-Mazz., A. rugulosus Maxim., and Turczaninovia fastigiata (Fisch.) DC. (i.e., A. fastigiatus Fisch.). Our second question is whether the correlation between narrowly lanceolate leaves and stream habitats in Aster is significant statistically?

In this study, we aim to (1) use three molecular markers to reconstruct the phylogeny of Aster to resolve the phylogenetic positions of the Aster species with narrowly lanceolate leaf and test the monophyly of the two similar species of Aster; (2) use Pagel’s trait evolution analysis methods (Pagel, 1994) to test whether the narrowly lanceolate leaf is significantly correlated with the riparian habitat, and if so to test the transformation ratio among four potential character combinations, and then to determine whether the riparian habitat drives the evolution of narrowly lanceolate leaf; and (3) examine the morphological and micro-morphological characters of the distinctive Aster species from Mt. Tongling and A. dolichophyllus to determine if the Tongling plant is a new species.

Materials and Methods

Taxon sampling

Seventy-three taxa were used for phylogenetic analysis, representing 19 related genera, the major clades of Aster, and one outgroup. Sequences of 71 of these species were downloaded from GenBank. Most species with narrowly lanceolate leaves in Aster were sampled. Five samples of the potential new taxon and six of the superficially similar Aster dolichophyllus were added in this study. The taxonomic treatment in the Flora of China and the definitions of Aster and “core Aster” in our previous study (Zhang et al., 2015) were followed. In the phylogenetic analysis, Chrysanthemum indicum L. was designated as outgroup as in previous studies (Li et al., 2012; Zhang et al., 2015). The ITS, ETS and trnL-F sequences were selected as molecular markers to generate the datasets. According to previous studies (Li et al., 2012; Zhang et al., 2015), the systematic position of shrub and alpine Aster groups are distinct from the core Aster (sensu Zhang et al., 2015; including the type of genus, Aster amellus) and may represent separate genera. Additionally, the habit of these two groups are significantly different from the core Aster. Therefore, in the character correlation analysis, a reduced data set was built to reconstruct the phylogeny of Aster. It was composed of the species above the clade of Aster nitidus Y. Ling and A. hersileoides Schneid. in Fig. 1. Two species, Aster nitidus and A. hersileoides, were set as roots according to our phylogeny and previous results (Li et al., 2012; Zhang et al., 2015). Voucher specimens for newly sequenced samples were deposited in PE. Voucher information and GenBank accession numbers are listed in Table S1.

Figure 1 Cladogram of the maximum likelihood (ML) phylogenetic tree of Aster.

Phylogenetic tree based on combined data (ITS, ETS and trnL-F), showing the position of Aster tonglingensis (in bold) and the species with narrowly lanceolate leaf (with green background). Values above branch represent bootstrap values (BS) and Bayesian posterior probabilities (PP), respectively; the dash (–) indicates BS < 50% or PP < 0.90.

DNA extraction, amplification, and sequencing

Leaf tissues were collected in the field and dried using silica gel. DNA extraction, purification, and sequencing followed the methods described by Zhang et al. (2015). Methods of PCR amplification of ITS and ETS sequences followed Linder et al. (2000), trnL-F sequence followed Zhang et al. (2015). The ITS primers of Linder et al. (2000), ETS primers “Ast-8” (Markos & Baldwin, 2001) and “18S-IGS” (Baldwin & Markos, 1998) and trnL-F primers “c” and “f” of Taberlet et al. (1991) were used.

Phylogenetic analysis

DNA sequences alignment was fulfilled using MAFFT online version (Katoh, Rozewicki & Yamada, 2017), and then was manually adjusted using BioEdit v7.0.8.0 (Hall, 1999). jModelTest 2.1.4 (Darriba et al., 2012) was used to select DNA substations module based on the Akaike information criterion (AIC). The GTR + G model was fit for ETS and ITS, and TVM + G model for trnL-F. Phylogenetic analyses were then conducted for two individual datasets, one consisting of ITS and ETS sequences, another consisting of trnL-F, and a combinative dataset. Phylogenetic trees were reconstructed using Maximum Likelihood methods and Bayesian Inference. Bootstrap support values (BS) for ML tree were calculated using 1,000 bootstrap replicates. Bayes inference was performed with 20 million generations, tree sampled every 1,000 generations. Bayesian posterior probabilities were calculated after omitting the first 500 trees (burn-in = 0.25). Analyses were done using RAxML 8.0.24 (Stamatakis, 2014) and MrBayes 3.2.4 (Ronquist et al., 2012) on the CIPRES science gateway portal (Miller, Pfeiffer & Schwartz, 2010). The parameter settings in Zhang et al. (2015) were followed.

Correlation evolution analysis

The Maximum likelihood and Bayesian methods for discrete character analyses (Pagel, 1994) were used. These analyses were accomplished in the program BayesTraits v 2.0 (Meade & Pagel, 2014). In order to reveal whether the leaf shape evolution and habitat are correlated, two traits including habitat (riparian versus non-riparian) and leaf shape (narrowly lanceolate (width/length < 0.15, see below) versus not narrowly lanceolate) were used to make the dataset. Habitat data were collected from floras (Chen, Brouillet & Semple, 2011), herbarium specimens (kept in PE), and our long-termed field observations in Eurasia. Due to the complexity of plant habitats, two definitions of riparian were used in the analysis (Table S2). One is a narrowly riparian habitat. All or the vast majority of individuals of species associated with narrowly riparian habitats occur only on the banks of rivers or streams. The other is a broadly riparian habitat. Species associated with this habitat occur not only on river or stream banks but also in other wet habitats (such as swamps and pool margins). The definition of leaf shape is based on the leaf shape index. It was obtained by dividing the length of each leaf by the width and then calculating the arithmetic mean of all specimens of each species. We measured the length and width of three middle cauline leaves of each specimen and for each species we measured ten specimens. These specimens were from PE, K, E, BM, and PRC herbaria. For the shrubby species, we measured the middle leaves of the first branch, as the main stem was usually leafless. For the species with a solitary capitulum, we measured lower leaves, as the middle part of the stem is leafless. In this study, leaves with a leaf shape index less than 0.15 were defined as narrowly lanceolate. This criterion approaches the traditionally recognized index (Stearn, 1983). Besides the narrowly lanceolate leaf shape, there were many other types of leaf shape that could be shaped by various factors. To eliminate the influence of these factors and to focus on the correlation between narrowly lanceolate leaf shape and riparian habitat, leaf index and habitat were treated as binary characters.

For the character correlation analysis, the DISCRETE module of BayesTraits v 2.0 (Meade & Pagel, 2014) that support binary characters was used to analyse the correlation of the two binary characters above (the first character is habitat, state 0 means non-riparian and state 1 means riparian; the second character is leaf shape index, state 0 means the index is more than 0.15 and state 1 means less than (including) 0.15; see Fig. 2). Two models were provided in this module, i.e., the dependent model and the independent model. We checked which model best fitted our data by comparing the maximum likelihood value obtained via the Maximum Likelihood (ML) method and the marginal likelihood value obtained via the Markov Chain Monte Carlo (MCMC) method. In the ML analyses, searching times for the maximum likelihood value of each calculation was set at 1000 and each calculation was repeated 10 times. In MCMC method analyses, priors were set as Gamma hyper-prior (Pagel, Meade & Barker, 2004) with default parameters. Marginal likelihood values were obtained by the stepping stones method (Xie et al., 2011). Based on the results of our preliminary analyses, the number of iterations was set at ten million with 100 stepping stones. Each calculation was repeated 10 times and then the final marginal likelihood value was obtained from the mean of ten marginal likelihood values. The parameter restriction test (Pagel, 1994) was used to determine the order of trait evolution. Each parameter (Fig. 2) was respectively set as zero in different runs to determine if any trait transition could be excluded from the process of trait evolution. Contingent change and temporal order test (Pagel, 1994) were employed to determine the dependence between the two traits and the acquisition order. The likelihood values of different analyses were compared using the likelihood ratio test (LRT, for ML results) and the Bayes Factors (BF, for MCMC results) test (Gilks, Richardson & Spiegelhalter, 1996) following the procedure recommended in the manual of BayesTraits v2 (Meade & Pagel, 2014).

Figure 2 Transitions among the four combinations of traits states.

The first trait is habitat, state 0 represents non-riparian habitat, state 1 represents riparian habitat; the second trait is leaf shape, state 0 represents leaf shape index > 0.15, state 1 represents leaf shape index ≤ 0.15. (A) non-narrowly lanceolate leaf in non-riparian habitat; (B) narrowly lanceolate leaf in non-riparian habitat; (C) non-narrowly lanceolate leaf in riparian habitat; (D) narrowly lanceolate leaf in riparian habitat.

Morphological and micro-morphological observations

For the description and the line drawings of the new species, living plants and herbarium specimens were examined by naked eye and under stereomicroscope. Living plants as well as FAA fixed materials were measured. The morphological comparison with other species of Aster was based on the study of herbarium specimens from PE (Chinese National Herbarium, Institute of Botany, Chinese Academy of Sciences).

The micro-morphological characters of the new species and its superficially similar species Aster dolichophyllus were examined. Voucher information of the materials is listed in Table S1. Anderson’s sectioning method (Anderson, 1954) was followed. For herbarium specimens, the capitula were stored in FAA solution for 24 h to soften tissues. The materials were then cleaned in a supersonic generator for 5 min at a frequency of 100 Hz. They were then transferred into a 5% NaOH solution and kept for 12 h for the study of anthers and 6 h for corolla and style. After cleaning with distilled water, the samples were transferred into a drop of Hoyer’s solution on microscope slides, and observed and photographed using a Leica DM5000B microscope. The corolla, filament collar, base and tip appendages of anthers, endothecial tissue, the stylopodium, stigmatic lines, and tip appendages of style were observed and measured under the light microscopy.

New taxon and the LSID statement

The electronic version of this article in Portable Document Format (PDF) will represent a published work according to the International Code of Nomenclature for algae, fungi, and plants (ICN), and hence the new names contained in the electronic version are effectively published under that Code from the electronic edition alone. In addition, new names contained in this work which have been issued with identifiers by IPNI will eventually be made available to the Global Names Index. The IPNI LSIDs can be resolved and the associated information viewed through any standard web browser by appending the LSID contained in this publication to the prefix “http://ipni.org/”. The online version of this work is archived and available from the following digital repositories: PeerJ, PubMed Central, and CLOCKSS.

Results

Phylogenetic results

When we compared the phylogenetic trees separately reconstructed based on the chloroplast and nuclear matrices, no obvious topology conflict was found. The two matrices therefore were combined in the following analyses. Consensus tree from BI analyses had nearly identical topologies with the ML tree. The best ML tree (−InL = 22036.02) is presented in Fig. 1. The topologies of our phylogenetic tree were largely consistent with previous studies (Li et al., 2012; Zhang et al., 2015). The species with narrowly lanceolate leaf were nested in different lineages on the tree (Fig. 1). Among them, Aster moupinensis formed a well-supported clade with two species without narrowly lanceolate leaves (A. smithianus Hand.-Mazz. and A. heterolepis Hand.-Mazz.) (Fig. 1, BS = 85, PP = 1.00) nested in the core Aster clade; A. sinoangustifolius Brouillet, Semple et Y.L. Chen lay at the base of the core Aster clade with strong support (Fig. 1, BS = 99, PP = 1.00); A. rugulosus was resolved as sister to A. scaber Thunb. (Fig. 1, BS = 100, PP = 1.00); Sheareria, the monotypic semi-aquatic genus, was placed in the core Aster clade, and resolved as sister to a clade consisting of the taxa from A. fanjingshanicus Y.L. Chen & D.J. Liu to A. souliei Franch. with moderate support (Fig. 1, BS = 78, PP <  0.90); Turczaninowia fastigiata formed a weakly supported clade (Fig. 1, BS = 67, PP < 0.90) with Aster procerus Hemsl.; Arctogeron gramineum (L.) DC.was resolved as sister to a clade consisting of the taxa from Asterothamnus fruticosus (C. Winkl.) Novopokr. to Aster poliothamnus Diels with moderate support (Fig. 1, BS = 67, PP = 0.94). Besides these species, the distinctive Aster species from Mt. Tongling (formally described as Aster tonglingensis below) and the similar species Aster dolichophyllus had similar narrowly lanceolate leaves. The phylogenetic results show that all individuals of Aster tonglingensis formed a strongly supported monophyletic clade (Fig. 1, BS = 100, PP = 1.00). It was nested in the strongly supported core Aster clade (Fig. 1, BS = 98, PP = 1), weakly resolved as sister to the subclade containing Aster tianmenshanensis G.J. Zhang and A. verticillatus (Reinw.) Brouillet, Semple & Y.L. Chen (Fig. 1, BS < 50, PP < 0.90). All individuals of A. dolichophyllus, formed another strongly supported clade nested outside of the core Aster (Fig. 1, BS = 100, PP = 1.00).

Correlation analysis of characters

Leaf shape indexes and the habitat information were kept in Table S2. For correlation analysis between leaf shape and broadly riparian habitat using ML method, the mean of the maximum likelihood value of the independent model was −33.85, that of the dependent model was −19.92, the likelihood ratio (LR) was 27.86, and the p-value of likelihood ratio (LRT) was smaller than 0.00001. For MCMC method, the mean of the log marginal likelihood value of the independent model was −36.85, the mean marginal likelihood value of the dependent model was −29.15, and the Log Bayes Factor was 15.40. For the correlation analysis between leaf shape and narrowly riparian habitat of ML method, the mean of the maximum likelihood value of the independent model was −29.70, that of the dependent model was −22.45, LR was 14.48, and the p-value of LRT was 0.0059. For MCMC method, the mean of the log marginal likelihood value of the independent model was −32.73, that of the dependent model was −28.84, and Log BF was 7.78. The detailed results of the analyses and the test of parameter restrictions are listed in Table S3.

Morphological and micro-morphological observation

Aster tonglingensis is similar to A. dolichophyllus in external morphology. They have similar narrowly lanceolate and leathery cauline leaves, and reflexed phyllaries (Figs. 3 and 4). But they are different in the shape of the basal leaves, leaf indument, bracteal leaves, and number of phyllaries series (Table 1). Aster tonglingensis has long petiolate and lanceolate basal leaves (Figs. 3F & 4A), a puberulent adaxial leaf surface (Fig. 4A), more than 30 capitula, single or several in terminal and axillary corymbs (Figs. 3E & 4A), whereas Aster dolichophyllus has spatulate and sessile basal leaves, a glabrous adaxial leaf surface, less than 10 capitula in a loose terminal corymb, and capitula never axillary. Aster tonglingensis has phyllary-like bracteal leaves and 5–7-seriate phyllaries (Fig. 3C), whereas A. dolichophyllus has bracteal leaves that are not phyllary-like and 2–3-seriate phyllaries. Aster tonglingensis differs from the closely related A. tianmenshanensis (Table 1) by its greater height (70–100 cm versus ca. 10 cm in A. tianmenshanensis), narrowly lanceolate leaves (versus spatulate) and more capitula (more than 30 versus only one), and differs from A. verticillatus (Table 1) by having large capitula (20–25 mm in diameter versus ca. 10 mm in A. verticillatus), beakless achenes (versus beaked in A. verticillatus), and a robust pappus (versus a readily caducous pappus in A. verticillatus).

Figure 3 Habitat and morphology of Aster tonglingensis.

(A) Aster tonglingensis growing in its riparian habitat; (B) inflorescence; (C) capitula and phyllaries; (D) disc florets; (E) cauline leaves and axillary capitula; (F) seedling.

Figure 4 Aster tonglingensis.

(A) habit; (B) capitula; (C) phyllaries; (D) bristle of pappus; (E) ray florets; (F) style branches of ray florets; (G) disc florets; (H) style branches of disc florets; (I) anthers.

Table 1 Morphological and micro-morphological characters of four Aster species.

Characters	Aster tonglingensis	Aster dolichophyllus	Aster verticillatus	Aster tianmenshanensis	
Height	70–100 cm	40–50 cm	25–100 cm	Up to 10 cm	
Basal leaves	Long petiolate, lanceolate	Sessile, spatulate	Petiolate, lanceolate	Sessile, spatulate	
Adaxial surface of leaves	Puberulent	Glabrous	Scabridulous	Glabrous	
Capitula	More than 30, terminal and axillary, 20–25 mm in diameter	Less than 10, terminal, 25–30 mm in diameter	More than 20, terminal and axillary, 10 mm in diameter	Single, terminal, 15–20 mm in diameter	
Phyllaries	5–7 series	2–3 series	3 series	2–3 series	
Achenes	Beakless	Beakless	Beaked	Beakless	
Pappus	Robust	Robust	Readily caducous	Robust	
Stigamatic lines	Equal to the sterile style tip appendages	Shorter than the sterile style tip appendages	Shorter than the sterile style tip appendages	Longer than the sterile style tip appendages	
Disc corolla lobes	Unequal, split to two thirds or three fourths of limb	Equal, split to one third of limb	Equal, split to one half of limb	Unequal, split to one third of limb	
Anther endothecial cells	Polarized thickened	Radially thickened	Radially thickened	Radially thickened	
Anther tip appendages	Narrowly triangular, length-width ratio ca. 2	Triangular, length-width ratio ca. 1.5	Triangular, length-width ratio ca. 1.5	Narrowly triangular, length-width ratio ca. 2	

In micro-morphological characters (Figs. 5 and 6), both Aster tonglingensis and A. dolichophyllus have lanceolate style branches (Figs. 5A & 6A), triangular style appendages (Figs. 5A & 6A), constricted style base (Figs. 5B & 6B), thickened filament collar (Figs. 5E & 6E), and obtuse and untailed anther base (Figs. 5E & 6E). However, A. tonglingensis differs from the latter (Table 1) by having long stigmatic lines equal to the length of the sterile style tip appendages (Fig. 5A) (versus shorter than the sterile style tip appendages in A. dolichophyllus, Fig. 6A), disc corolla lobes split to two thirds or three fourths of the limb of the disc floret corolla (Fig. 5C) (versus split to one third in A. dolichophyllus, Fig. 6C), narrowly triangular anther tip appendages with length-width ratio ca. 2 (Fig. 5D) (versus triangular with length-width ratio ca. 1.5 in A. dolichophyllus, Fig. 6D), and a majority of anther endothecial cells polarized thickened (Fig. 5F) (versus radially thickened in A. dolichophyllus, Fig. 6F).

Figure 5 Micro-morphology of Aster tonglingensis.

(A) Style branches; (B) stylopodium; (C) corolla; (D) anther tip appendage; (E) anther base appendage, filament collar and anther endothecial tissue.

Figure 6 Micro-morphology of Aster dolichophyllus.

(A) Style branches; (B) stylopodium; (C) corolla; (D) anther tip appendage; (E) anther base appendage, filament collar and anther endothecial tissue.

Aster tonglingensis is also different from its related species morphologically (Table 1). It differs from A. tianmenshanensis (Zhang et al., 2015) by having stigmatic lines as long as the sterile style tip appendages (Fig. 5A) (versus only one third as long as the appendages in A. tianmenshanensis), disc corolla lobes split to two thirds or three quarters of the limb of the disc floret corolla (Fig. 5C) (versus half way in A. tianmenshanensis), narrowly triangular anther tip appendages with length-width ratio ca. 2 (Fig. 5D) (versus triangular with length-width ratio ca. 1.5 in A. tianmenshanensis), and a majority of anther endothecial cells polarized thickened (Fig. 5F) (versus radially thickened in A. tianmenshanensis). It differs from A. verticillatus by the latter having stigmatic lines two times longer then appendages, disc lobes split to half the length of the limb, and the majority of anther endothecial cells radially thickened (Zhang et al., 2015).

Discussion

Convergent evolution of the narrowly lanceolate leaf in the genus Aster

In our study, the traditionally defined Aster (Ling, Chen & Shih, 1985; Nesom, 1994; Chen, Brouillet & Semple, 2011) was not a monophyletic group. Some genera like Asterothamnus, Rhinactinidia, Arctogeron, and Myriactis were nested within different clades of Aster and formed a weakly supported clade (BS < 50, PP < 0.9) with members of the traditionally defined Aster. This result was congruent with previous studies (Li et al., 2012; Zhang et al., 2015). Our molecular phylogenetic analysis indicated that the species with narrowly lanceolate leaves were nested in distantly related lineages of the genus Aster, implying that they originated independently at least eight times (Fig. 1). Narrowly lanceolate leaves are the results of convergent evolution in the genus Aster.

Correlation evolution between leaf shape and habitat

In our analysis, for broadly riparian habitat, p-value of LRT of ML method between two models was smaller than 0.01 and Bayes Factor of MCMC method was 15.40. These results suggested that the riparian habitat and narrowly lanceolate leaf shape was strongly correlated. The results of ML method parameter restriction tests showed all parameter could be exclusive (no significant difference with zero). This means every character state transitions were possible. When we set the opposite transition rates as equal, there was no significant difference comparing to unequal. This results showed that the order of character state transition was not clear. The single parameter test with MCMC method showed that parameter q24 was strongly supported to differ from zero (BF = 5.34) and parameters q21 (BF = 2.05) and q34 (BF = 3.08) were positively supported to differ from zero. Other parameters did not differ from 0 in these tests. To set the opposite parameters as equal, the results positively supported q13 is not equal to q24 (BF = 3.01). The dependent model test showed q13 (7.08) was much smaller than q24 (62.90). These results indicated that, compared to the plants with wide leaves, the plants with narrow leaves tended to transfer more frequently to a riparian habitat. But our temporal order test showed that no significant order could be recognized. Therefore, we could not determine whether the riparian habitat or narrowly lanceolate leaves came first in this adaptive process.

Some broadly riparian species are not strictly growing on stream banks. So we did the same test for narrowly riparian habitat species. Most results are similar to the analysis for broadly riparian habitat. In ML method analysis, the likelihood ratio of two models is 14.48 with p-value less than 0.01. The Bayesian factor of MCMC method is 7.78. These results showed that the riparian habitat was strongly correlated with narrowly lanceolate leaves. However, the single parameter test with ML method showed that no parameter was strongly supported to differ from zero. The order of character states could not be fixed. The MCMC method parameter restriction tests showed that q12 (BF = 7.11) and q24 (BF = 6.25) were strongly supported to differ from zero, q21 (BF = 2.41) and q34 (BF = 2.56) are positively supported to differ from zero. Other parameters were not supported as different from zero. These results showed that the transitions from wide leaves to narrowly lanceolate leaves in non-riparian habitat and from non-riparian habitat to riparian habitat with narrow leaves cannot be ignored. These transitions indicated the potential path from non-riparian with wide leaves to riparian with narrow leaves in genus Aster. Furthermore, when setting q13 = q24, the negative result was supported. This result showed that the habitat changed from non-riparian to riparian likely depending on the narrowly lanceolate leaf shape. The test positively supported the q34 differs from zero. We also could find that q34 (= 54.63) were much large than q12 (= 12.06). These results suggested that the habitat was more likely to change from non-riparian to riparian when the plants have narrowly lanceolate leaves. Our directivity test showed that the q12 was not significantly different from q13. But q12 (12.06) was larger than q13 (7.08). Furthermore, q12 was proved significantly different from zero whereas q13 was not. This showed the rate of transition from wide leaf in non-riparian habit to narrowly lanceolate leaf in non-riparian habitat was higher than the rate of transition from wide leaf in non-riparian habitat to wide leaf in riparian habitat. Based on the results above, we propose that the narrowly lanceolate leaves trait acquisition was likely earlier than the riparian habitat acquisition in these riparian species with narrowly lanceolate leaves.

In our analysis, in both the broadly riparian species and the narrowly riparian species of Aster, habitat was strongly correlated with leaf shape (p-value < 0.00001 and BF = 15.40 for broadly riparian habitat; p-value < 0.01 and BF = 7.78 for narrowly riparian habitat). Our MCMC test supported that plants with narrowly lanceolate leaves were more likely to change to riparian habitat than those with wide leaves (q24 much large than q13, see Table S3). Our directivity analysis showed that some Aster species may have effectively employed a preadaptation strategy (Shelley, 1999; Kangas, 2004; Losos, 2013), i.e., developing narrowly lanceolate leaves first, and then adapting to the riparian habitat. This pre-adaptation strategy could reduce the risk when plants encountered new habitats by chance (Shelley, 1999; Kangas, 2004; Losos, 2013). As shown in the case of Ainsliaea (Mitsui et al., 2011), individuals with broader leaves could be swept away under strong selection pressure within the species (e.g., damage to the broader leaves by strong water flow), while members with narrowly lanceolate leaves could survive. But the majority of our single parameter tests was not strongly supported (p-value > 0.01 and BF < 2), suggesting that the process of adaptation to the riparian habitat in genus Aster may be very complicated. Other factors could also contribute to the formation of narrowly lanceolate leaves. For instance, Arctogeron gramineum, a species having narrowly lanceolate leaves grows in extremely dry habitat instead of riparian habitats. Thus, although our results supported the strong correlation between narrowly lanceolate leaves and riparian habitat in the genus Aster, the details of the biological connections between them could be complicated.

To sum up, the riparian habitat and narrowly lanceolate leaf shape were strongly correlated in the genus Aster based on our comparative analysis. Some test results suggested that the pre-adaption strategy could be an important factor in the adaptation of the Aster species to the riparian habitat.

Aster tonglingensis as a new species: evidence from morphology, micromorphology and molecular phylogeny

In the phylogenetic tree, all individuals of A. tonglingensis formed a strongly supported clade nested in the core Aster clade (Fig. 1). All the accessions of the similar A. dolichophyllus formed a strongly supported clade outside this clade (Fig. 1). In summary, our molecular analysis indicated that A. tonglingensis was a strongly supported monophyletic group and a unique lineage quite distinct from the lineage including A. dolichophyllus (Fig. 1).

Although they look similar, Aster tonglingensis and A. dolichophyllus are different in many characters. Both of them do have similarly shaped narrowly lanceolate, leathery cauline leaves, reflexed phyllaries, and almost glabrous leaf surfaces (Figs. 3 and 4). However, the shape of their basal leaves is totally different. The basal leaves of A. tonglingensis are lanceolate with a long petiole (Figs. 3F & 4A), whereas those of A. dolichophyllus are spatulate and sessile. In the previous studies of Aster (Ling, Chen & Shih, 1985; Nesom, 1994; Chen, Brouillet & Semple, 2011) based on herbarium specimens, many lacked descriptions of the basal leaves because in many species these were withered by the time of flowering. The present study showed that the character of the basal leaves can be taxonomically important. In addition, many floral characters of these two species are different. For instance, A. tonglingensis differs from A. dolichophyllus by having terminal and axillary corymbs and more than 30 capitula (Figs. 3 and 4), whereas the latter species has a lax terminal corymb with usually fewer than 10 capitula. Axillary corymbs (Fig. 3E) are rare in Eurasian Aster, such as A. turbinatus and A. verticillatus. Both Aster tonglingensis and A. dolichophyllus have reflexed phyllaries, but the 5–7-seriate phyllaries and the enormous linear bracteal leaves below the involucres of A. tonglingensis (Fig. 3C) show the obvious differences in comparison with the 2–3-seriate phyllaries and few lanceolate bracteal leaves of A. dolichophyllus. Aster tonglingensis has longer disc corolla lobes, about two thirds as long as the limb of the floret (Fig. 5C), whereas the lobes of A. dolichophyllus are only one third as long as the limb. Furthermore, the disc florets of A. tonglingensis (ca. 5–7 mm in length, Fig. 4G) are significantly smaller than those of A. dolichophyllus (ca. 9–11 mm in length). These differences of floret characters may be related to their pollination.

Our molecular analysis shows that the most closely related species of Aster tonglingensis are A. tianmenshanensis and A. verticillatus. However, both of them differ from A. tonglingensis in their gross morphology: A. tianmenshanensis is a small herb with a solitary capitulum and spatulate leaf blades growing on limestone cliffs. A. verticillatus has tiny capitula (involucre ca. 2–7 mm in diameter), and beaked achenes with a readily caducous pappus.

Micro-morphological characters are useful in the taxonomy of Aster. For instance, Zhang et al. (2015) suggested the morphology of the stigmatic lines was an important character to distinguish different species. Similarly, the stigmatic lines could distinguish A. tonglingensis from the externally similar species as well as the closely related species. The stigmatic lines of A. tonglingensis are as long as the sterile style tip appendages (Fig. 5A), while those of A. dolichophyllus are two times longer than the sterile style tip appendages, those of A. tianmenshanensis are one third as long, and those of A. verticillatus are three times as long. This character is expected to be related to their pollination. In addition, the anther endothecial cells and shape of sterile anther tip appendages are also different in these species (Fig. 5) (also see Zhang et al., 2015).

To sum up, Aster tonglingensis is different from A. dolichophyllus in both gross morphological and micro-morphological characters. Therefore, A. tonglingensis is a unique new taxonomic entity. Moreover, our molecular phylogenetic analysis showed that A. tonglingensis is a unique lineage (Fig. 1). Therefore, we treated it as a new species. The similar narrowly lanceolate leaves shared by A. tonglingensis and A. dolichophyllus are probably the result of convergent evolution.

Taxonomic treatment

Aster tonglingensis G.J. Zhang & T.G. Gao, sp. nov. Fig. 4.

Type: CHINA. Zhejiang Province, Wencheng County, Mt. Tongling, elev. 640 m, 119°52′E, 27°49′N, 2nd Sept. 2013, H. H. Hu 331-1 (holotype PE!, isotype PE!).

Diagnosis: The new species superficially resembles Aster dolichophyllus Ling. Both species have narrowly lanceolate cauline leaves, recurved phyllary tips, and occur near streams. However, the phyllaries of Aster tonglingensis were 5–7-seriate, green (vs. 2–3-seriate, green with purple tip in A. dolichophyllus), capitula usually more than 30, both terminal and axillary (vs. less than 10, only terminal), adaxial surface of all leaves puberulent (vs. glabrous), basal leaves lanceolate, apex rounded or obtuse (vs. spatulate, apex acute), corolla of disc floret 5–7 mm, lobes half to two thirds as long as limb (vs. corolla 9–11 mm, lobes one third as long as limb), pappus whitish (vs. slightly brown).

Perennial herb. Rhizomes thin, transverse, slightly woody, 3–15 cm long, ca. 0.3–0.5 cm in diameter, sometimes expanded near the base of stem becoming a 2–4 cm hard node. Stem solitary or two to three together, erect, 70–100 cm high (including inflorescence), unbranched except for inflorescence, lower part glabrous, upper part puberulous, leafy. Leaves of rosette lanceolate, 4–18 × 0.8–2.5 cm, base gradually narrowing, margin serrately four to eight toothed, petiole 3–10 cm long, apex acute; lower cauline leaves similar to rosette leaves, sessile or petiole obscure, narrowly lanceolate, 4–13 × 0.4–1 cm, margin entire or serrately 3–5-toothed, base gradually narrowing, apex acute; all leaves thinly leathery, abaxially glabrous and light green, main vein and lateral veins prominent, adaxially puberulent, dark green and glossy. Capitula usually more than 30, in one to five terminal and axillary corymbs, peduncle puberulous, with dense bracteal leaves, bracteal leaves ciliate, abaxially glabrous, adaxially densely puberulous; involucre campanulate, ca. 8–10 mm long, 5–8 mm in diameter, phyllaries in 5–7 imbricate series, green, lanceolate, 5–7 × 1–1.5 mm, hardened at their bases, herbaceous above, the outer shorter than the inner, ciliate, upper part of abaxial surface densely puberulous, with a revolute acute apex, ca. 1 mm long, both surfaces densely puberulous. Ray florets ca. 15, female, with a greenish, glabrous tube ca. 3 mm long; ligules whitish, lanceolate 7–10 × ca. 2 mm, with four nerves, apex with two or three teeth. Disc florets many, hermaphrodite, corolla greenish white to yellow, tube greenish and puberulent at the top, ca. 3 mm long, thin but expended at base, lobes five, lanceolate, unequal, two thirds as long as limb. Achenes of both florets identical, narrowly oblong, four-ribbed, ca. 2 mm long, puberulous, lower part densely so; pappus uniseriate, whitish, bristles barbellate, ca. 7 mm long, nearly as long as disc corolla at anthesis. Flowering in July.

Etymology: The species is named after its type locality, Mt. Tongling, Wencheng County, Zhejiang Province, China.

Conservation status: Aster tonglingensis is a very narrowly distributed species and is currently known only from one stream in Mt. Tongling Natural Reserve. A population with ca. 100 individuals was found along the stream. We scoured nearby places with similar habitats in this region but failed to find more populations. This part of the natural reserve currently is open to tourists. A footpath was built along this stream which passes through its location. The habitat of A. tonglingensis is easily disturbed or damaged. According to Criteria B2a of International Union for Conservation of Nature Red List Categories, this species should be treated as Critically Endangered. More attention and protection should be paid to this new but vulnerable species.

Additional specimens examined (paratypes): CHINA. Zhejiang province, Wencheng county, Mt. Tongling, elev. 640 m, 119°52′E, 27°49′N, 2nd Sept. 2013, H. H. Hu 331-2, 331-3, 331-4 & 331-5 (PE !).

Conclusions

Leaf shape has been used as an important character in the taxonomy of Aster for a long time (Chen, Brouillet & Semple, 2011). The relationship between it and the environment, however, has never been investigated. In the present study, a phylogeny including most species with narrowly lanceolate leaf in Aster was reconstruction based on three molecular markers. It was revealed that species with narrowly lanceolate leaves were placed in far related lineages of the genus Aster (Fig. 1). Thus, the narrowly lanceolate leaf shape originated independently several times in the genus Aster. It was the result of convergent evolution. Comparative analysis in the phylogenetic context revealed that narrowly lanceolate leaf shape and riparian habitat were strongly correlated. The transition order of riparian habitat and narrowly lanceolate leaf was shown to be usually uncertain. But the preadaptation of the narrowly lanceolate leaf was positively supported by some analysis (Fig. 2). In summary, convergent evolution and preadaptation may play important roles in the evolution of leaf shape in the genus Aster. Meanwhile, an unexpected new species with narrowly lanceolate leaves, Aster tonglingensis, was discovered and established based on the evidence of molecular, morphology and micro-morphology. This new species was descripted and illustrated here.

Asteraceae is the largest and relatively young plant family (Funk et al., 2009; Heywood, 2009). Simultaneously, members of this mega-diverse family show abundant morphological diversity (Funk et al., 2009). They occur in almost every corner of the earth and occupy various habitats (Funk et al., 2009), thus providing an excellent opportunity to study convergent evolution (Heywood, 2009). The present study provided new insights into the process of convergent evolution of leaf form in a big genus of this mega-diverse family. In turn, understanding more details of the convergent evolution in this family helped to discover the cryptic biodiversity before they go extinct, as shown in the unexpected discovery of the new species Aster tonglingensis in this study.

Supplemental Information

Table S1 GenBank accession numbers of the ITS, ETS and trnL-F sequences of the sampled taxa used in this study

Click here for additional data file.

Table S2 Data of the leaves used in traits evolution analysis

Click here for additional data file.

Table S3 Detailed results of the traits evolution analysis

Click here for additional data file.

We are grateful to Mr. Yunxi Zhu for his illustration. We thank the curators of herbaria PE, K, E, BM, and PRC who granted us access to their collections and photos, and the staff from Mt. Tongling Natural Reserve for their help in the field work.

Additional Information and Declarations

Competing Interests

Author Contributions

DNA Deposition

Data Availability

New Species Registration

The authors declare there are no competing interests.

Guo-Jin Zhang performed the experiments, analyzed the data, contributed reagents/materials/analysis tools, prepared figures and/or tables, authored or reviewed drafts of the paper, approved the final draft.

Hai-Hua Hu performed the experiments, contributed reagents/materials/analysis tools, prepared figures and/or tables.

Tian-Gang Gao conceived and designed the experiments, contributed reagents/materials/analysis tools, prepared figures and/or tables, authored or reviewed drafts of the paper, approved the final draft.

Michael G Gilbert authored or reviewed drafts of the paper, approved the final draft.

Xiao-Feng Jin contributed reagents/materials/analysis tools.

The following information was supplied regarding the deposition of DNA sequences:

The GenBank accession numbers of newly sequenced samples are available in Table S1.

The following information was supplied regarding data availability:

The raw measurements of leaf shape and the habitat information are available in Table S2.

The following information was supplied regarding the registration of a newly described species:

Aster tonglingensis G. J. Zhang et T. G. Gao LSID: 77192771-1.

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
