# Peer review of "Convergent origin of the narrowly lanceolate leaf in the genus Aster—with special reference to an unexpected discovery of a new Aster species from East China"

_PeerJ, doi:10.7717/peerj.6288_

## Round 0.1 · original submission · Minor Revisions

The reviewers and I are in agreement that this manuscript is worthy of publication after minor revision. Please note the annotations made by the reviewers. You do not need to make all the marked alterations but should explain why, if you have chosen not to follow the advice.

Reviewer 1 ·

Basic reporting

The work provides sufficient context for the reader to engage with the topic, and the references given are relevant and necessary. The hypothesis is linked to the methodology used and made the focus of the discussion and species description. The specimen photographs as well as the botanical diagram are helpful and with appropriate scale bars. The species description at the end contains necessary detail. Some proof reading would help pick up on errors and typos in the manuscript.

Experimental design

As my background is not molecular, I will focus on my area of expertise which is morphology. The use of rudimentary linear morphometric tools is sufficient to answer the questions posed. The analysis of leaf shape and habitat correlation was interesting although I wonder why the authors elected to use a technique that required categorisation of the leaf shape index? The authors clearly state that this is in line with the published knowledge but I think this paragraph (line 165) would benefit by a further explanatory statement on the use of a discrete method as opposed to a continuous. Further, is there a bias from the use of a ratio as opposed to inspecting covariance of the measured variables independently?

Validity of the findings

In the results presentation, the morphological species comparison would benefit for a comparison table, for example the findings in paragraph starting line 277 would be easier to follow in a table format.

·

Basic reporting

The paper is generally well-written. Nonetheless, I made some suggestions on the manuscript to help improve the text.
The introduction provide a clear context for the study. The literature cited is relevant and appears thorough.
The structure of the paper is conform to the norms in the field.
The figures are pertinent and sufficient for the purpose of the paper. In particular, the photographs and line drawing of the new taxon are excellent.
I checked the authors' deposit statements and all seems in order. The GenBank data are already available and were properly deposited.

Experimental design

The research reported is novel and is well within the scope of the Journal. Nobody has reported on leaf character evolution in tribe Astereae (Asteraceae) before and the subject is interesting also in a wider context.
The research questions raised by the authors are pertinent and appropriately defined given context provided.
The methods reported for plant phylogenetic analysis are standard in the field and are well-known and proven. The taxonomic sampling appears adequate given the aims of the study.
The Pagel approach used for correlations is appropriate for the type of question raised by the authors.

Validity of the findings

The authors could have confined themselves to merely indicate that evolution of narrowly lanceolate leaves result from parallel evolution in Aster (a result readily discernible from the phylogeny presented), but they also investigated a possible driving force behind the evolution of that trail: its correlation with riparian habitats. They could not identify however, a precise order of changes. Based on their analyses, the authors suggest a most likely path, nonetheless. This appears reasonable.
Taxonomic novelty: the authors have conformed with PeerJ policies concerning the new species and meet ICBN standards.
The authors clearly demonstrate the distinction between their new taxon and the superficially similar A. dolichophyllus, both phylogenetically and micromorphologically.
The taxonomic description at the end of the paper is of high quality and thorough.
The overall conclusions are cogent.

Additional comments

This is an interesting paper that combined taxonomy and evolutionary biology, each supporting the other. It is infrequent that this is done in plant taxonomy and the authors are commanded for having done so.

---

## Round 0.2 · accepted · Accept

Thank you for dealing so comprehensively with the feedback from the reviewers.

# Reviewer 1 ·

Basic reporting

No further amendments required.

Experimental design

No further amendments required.

Validity of the findings

No further amendments required.

Additional comments

The amended manuscript has greatly improved in both language and clarity, after incorporating the comments provided by the reviewers. The current text reads well, with sufficient explanatory information included. The added comparison table helps the reader compare the four taxa easily. I recommend this manuscript for publication.